

# The deployment of temporary nurses and its association with permanently-employed nurses' outcomes in psychiatric hospitals: a secondary analysis

Leonel Oliveira[1], Beatrice Gehri[1,2] and Michael Simon[1]

[1] Institute of Nursing Science, Department of Public Health, University of Basel, Basel, Switzerland
[2] University Psychiatric Clinics, Basel, Switzerland

## ABSTRACT

**Aims:** This study's objective was to investigate possible associations between the frequency of temporary nurse deployments and permanently-employed nurses' outcomes including staffing levels in Swiss psychiatric hospitals.

**Background:** Faced with widespread nursing shortages, some nursing managers frequently deploy temporary nurses to meet their staffing needs. While various studies have investigated the relationships between temporary nurses' deployment and permanently-employed nurse outcomes, few anywhere, and none in Switzerland, have explored such deployments' relationships with permanently-employed nurses' job satisfaction, burnout, or intent to leave their organization or profession. Furthermore, especially in psychiatric hospitals, research on temporary nurse deployments and their association with permanently-employed nurses' outcomes remains scarce.

**Methods:** This secondary analysis is based on the Match[RN] Psychiatry study, which included 79 psychiatric units and 651 nurses. Using descriptive analyses and linear mixed modeling, we assessed the frequency of temporary nurses' deployment and its association with four permanently-employed nurse outcomes: staffing levels, job satisfaction, burnout, and intention to leave their organization or profession.

**Results:** Roughly one-quarter of the studied units reported frequently deploying temporary nurses. Nonetheless, no differences in nurse staffing levels were found. Regarding permanently-employed nurses' outcomes, we identified slightly higher levels of intention to leave the profession (beta = 0.18; 95% CI [0.03–0.33]) and burnout (beta = 0.19; 95% CI [0.4–0.33]) on units where temporary nurses were frequently deployed.

**Conclusion:** Deploying temporary nurses appears to help units maintain adequate staffing levels. However, additional research will be necessary to better understand whether working conditions are the common cause of temporary nurses' deployment and permanently-employed nurse outcomes. Until more information is available, unit managers should consider alternatives to deploying temporary nurses.

Corresponding author
Michael Simon, m.simon@unibas.ch

## INTRODUCTION

In recent years an aging population, an increase in retiring nurses, a lack of new nurses and the inability to retain nurses in the profession have led to a growing nursing shortage (*Michel & Ecarnot, 2020*). All of these factors complicate unit managers' efforts to bridge short-term gaps opened up by vacant positions, absences, and sick leave. Therefore, especially where necessary working hours cannot be supplied by nurses from their own or other units, managers commonly consider bringing in temporary nurses (also known as supplementary nurses) (*Hurst & Smith, 2011*).

Temporary nurses are hired to bolster nurse staffing and maintain staffing levels (*Aiken et al., 2007*; *Hurst & Smith, 2011*). There are two broad classes of temporary nurse: internal and external (*Bae et al., 2015*). Internal temporary nurses, who are also referred to as bank nurses, are nurses employed directly by an institution to carry out short-term assignments on different units (*Hurst & Smith, 2011*). The duration of their assignments vary substantially—from a single shift up to several months. External temporary nurses, often referred to as agency nurses, are employed by outside agencies. As with their internal counterparts, these nurses are requested based on demand, for a similar range of assignment durations (*Aiken et al., 2007*).

Although the frequency and extent to which temporary nurses are used varies greatly, studies in the United States (US) have found that around three-quarters of hospitals commonly employ them. On average, 9% of all hospital nursing staff are temporarily deployed (*Aiken et al., 2007*; *May, Bazzoli & Gerland, 2006*; *Staggs & He, 2013*). Further investigations of US hiring practices indicated that roughly half of temporary nurses hired there are external (*Bae, Mark & Fried, 2010*).

On average, an external temporary nurse costs $21 per hour more than a permanently-employed one (*Xue et al., 2015*). A cost analysis conducted in England found that in 2007 and 2008, 5–52% of hospital expenditures for nursing staff there were spent on temporary nursing staff (*Hurst & Smith, 2011*). However, in cases of understaffing, the costs of temporary nurses seem not to exceed those for overtime worked by permanently-employed nurses (*Faller, 2019*).

Regarding understaffing, (*Shin, Park & Bae, 2018*) meta-analysis showed correlations between increased patient-to-nurse ratios and the probability of burnout, poor job satisfaction and intention to leave. As temporary nurses reduce understaffing, they may help prevent or reduce such negative outcomes.

In spite of their contribution to staffing, evidence supporting temporary nurses' effects on permanently-employed nurse's outcomes remains scarce. However, certain negative outcomes have surfaced. For example, temporary nurses' insecurities regarding hospital infrastructure and a lack of orientation in new work environments increases workload and stress for permanently-employed nurses, while understaffing increases the volume of necessary care left undone (*Berg Jansson & Engström, 2017*). Still, frequent deployment of temporary nurses has never been associated with job dissatisfaction, intention to leave the organization, or burnout among permanently-employed nurses (*Aiken et al., 2007*). On the contrary, engaging them may have a net positive impact on spreading knowledge and

prompting change: they distribute the experience and knowledge they gain from each new assignment through other units and hospitals (*Berg Jansson & Engström, 2017*).

To date, studies investigating the effects of temporary nurses' deployment have focused almost exclusively on acute somatic hospitals. Only one study analyzed temporary nurses in psychiatric hospitals, observing that the patients' seclusion increased with higher deployments of temporary nurses (*Yurtbasi et al., 2021*). Psychiatric hospitals differ substantially from somatic hospitals regarding the scope of their activity, their patient populations, and for their strong focus on risk assessment, especially for suicidality or deescalating interventions. And while psychiatric nurses tend to rate their nurse-doctor relationships and staffing adequacy on their units higher than their colleagues in acute-care hospitals, they give comparatively low ratings for participation in hospital affairs, the foundations of nursing quality, further training opportunities and leadership (*Roche & Duffield, 2010*). Given the identified differences in the work environment in psychiatric and somatic hospitals it is vital to understand the relationships between temporary nurse deployment and permanently-employed nurses' outcomes in the psychiatric setting.

In summary, most studies regarding temporary nurses have been conducted in acute somatic hospitals; to our knowledge, few have examined the prevalence of temporary nurses' deployment and its associations with permanently-employed nurses' outcomes; and no such association has been studied in psychiatric hospitals. We know that temporary nurses are deployed in order to maintain adequate nurse staffing levels—defined as the appropriate number of nurses to provide care for patients meeting all reasonable requirements in the relevant situation (*Welsh Government, 2016*)—but it has not been investigated whether those levels are ultimately achieved. To date, studies have shown that understaffing has a negative impact on nurse outcomes, while frequent deployment of temporary nurses has not been associated with poorer outcomes in permanently-employed nurses.

Therefore, this study had three objectives: (1) to determine the prevalence of temporary nurses' deployment; (2) to assess whether units frequently deploying temporary nurses have nurse staffing levels similar to those that never or less frequently use temporary nurses; and (3) to explore associations between the deployment of temporary nurses and permanently-employed nurses' outcomes, namely job satisfaction, burnout, intention to leave their organization and intention to leave their profession in Swiss psychiatric hospitals.

## MATERIALS AND METHODS

### Design and setting

This study is a secondary analysis of the cross-sectional multicenter "Matching Registered Nurse services with changing care demands in psychiatric hospitals" (Match[RN] Psychiatry) observational study (*Gehri et al., 2021*). That study was conducted between September 2019 and June 2021 in psychiatric hospitals treating adult inpatients in the German-speaking region of Switzerland. Of 40 psychiatric hospitals with membership in the Association of Nursing Directors of Psychiatric Institutions in Switzerland

("Vereinigung Pflegekader Psychiatrie Schweiz"; VPPS), thirteen agreed to participate in the Match^RN Psychiatry study (*Gehri et al., 2021*).

The original study was reviewed by the responsible ethics committee (Ethics Committee Northwestern and Central Switzerland). The nurse questionnaire contained a cover letter describing the purpose of the study, the anonymity of the data collection and data protection measures. Participation was voluntary and considered as consent. As the survey was anonymous the study received an unrestricted waiver on 2019-09-27 as the study is outside the ethics committee's jurisdiction (Project ID: Req-2019-00589) (*Gehri et al., 2021*).

## Data source and sample

The Match^RN Psychiatry study data were compiled in two sets: one for unit-level data, one for nurse-level data.

Unit-level data included questionnaires completed by unit managers, with 57 items assessing unit staffing and the deployment of temporary nurses, as well as planning principles and options. Nurse-level data are drawn from a 164-item questionnaire on structural factors (*e.g.*, the number of nurses present on the last shift and the number of patients cared for), characteristics of the nurses' work environment, safety culture, work processes, and sociodemographics, along with other nurse outcomes, *i.e.*, job satisfaction, well-being, experiences with patient violence against nurses, and professional experience in nursing. Data from nurses working on adult units selected by the hospitals themselves were collected using two available methods, web-based or paper-pencil questionnaires. A single point of contact person coordinated and managed the onsite data collection.

The original Match^RN Psychiatry study data included 115 units from 13 Swiss psychiatric hospitals. The surveys received responses from 1,185 nurses (overall response rate of 71.5%, range per hospital: 51–88%). Our analysis only used data from units that answered the questions on temporary nurses' deployment. Our nurse data set was limited to registered nurses who completed the items of interest to our study. Details regarding data collection and methodology can be obtained from the corresponding study protocol (*Gehri et al., 2021*).

## Variables and measures
### Outcome variables

Job satisfaction was measured using data from one RN4CAST Likert-style survey question (*Sermeus et al., 2011*): "How satisfied are you with your organization, considering all aspects?" The ordinal answer options ranged from 1 ("Very dissatisfied") to 4 ("Very satisfied"). Intention to leave the organization was measured using the question "How often during the course of the past year have you thought about looking for a new job?" In this case, there were five answer options: 1 ("Never"); 2 ("Several times a year"); 3 ("Several times a month"); 4 ("Several times a week"); and 5 ("Daily"). Intention to leave the profession was asked with the question "How often during the course of the past year have you thought about leaving your profession of nursing?" This used the same five answer options as the item on intention to leave the organization. Both questions were drawn from

the NEXT survey (*Hasselhorn & Müller, 2005*). Emotional exhaustion was assessed using five questions from the Copenhagen Psychosocial Questionnaire (COPSOQ) (*Kristensen et al., 2005*): (1) "How often are you physically exhausted?"; (2) "How often are you emotionally exhausted?"; (3) "How often do you feel exhausted?"; (4) "How often do you come to work even though you feel very sick and uncomfortable?"; and (5) "How often are you unable to leave work behind in your free time?" Ordinal answer options ranged from 1 ("Always") to 5 ("Almost never"). We coded the answer options for burnout in reverse order and summarized them as a combined mean rating. Cronbach's alpha for the burnout data was acceptable at 0.75 (95% CI [0.72–0.78]). The authors have permission to use the instruments from the copyright holders.

### Predictor variables

From the unit-level data, we assessed the frequency of deployment of temporary nurses using a single item: "How often are temporary nurses used on your unit?" This had six ordinal scaled answer options: (0, "Never"; 1, "1–4 times a year"; 2, "5–10 times a year"; 3, "Once a month"; 4, "Several times a month"; and 5, "Several times a week." For our analysis, we dichotomized the responses as 1 for "Occasionally or never" (*i.e.*, responses 0–2 "Never" to "5–10 times a year"), and 2 for "Frequently" (*i.e.*, responses 3–5, "Once a month" to "Several times a week").

From the nurse data set we included three sociodemographic variables: (1) age (in years); (2) gender (1 = "Female"; 2 = "Male"); and (3) employment percentage (in percentage). For employment percentage nurses were asked whether they worked in their hospitals full-time or part-time; if part-time, they were asked for details on their employment percentage. We assigned the results to three categories: (1) <60%; (2) 61–95%; and (3) 96–100%.

To gauge the nurses' perception of their units' leadership, we used the four-item "Nurse Manager Ability, Leadership, and Support of Nurses" sub-scale of the "The Practice Environment Scale of the Nursing Work Index (PES-NWI)" (*Lake, 2002*). Each item included four Likert-type answer options, from 1 ("Disagree") to 4 ("Agree"). There were four questions in this subscale: (1) "A nurse manager that is supportive of the nurses."; (2) "A nurse manager who is a good manager and leader."; (3) "A nurse manager who praises and recognizes a job well done."; and (4) "A nurse manager who backs up the nursing staff in decision-making, even if the conflict is with a physician.". We summarized the four items by their combined mean rating. Cronbach's alpha showed good internal consistency for leadership data with 0.83 (95% CI [0.81–0.85]). The authors have permission to use this instrument from the copyright holders.

To measure nurse staffing levels ('adjusted staffing'), we followed *Bachnick et al.'s (2018)* staffing approach. We first calculated the patient-to-nurse ratio by dividing the total number of patients by the total number of registered nurses reported in the nurse survey for the most recent shift. We then adjusted the calculated patient-to-nurse ratio using a mixed-effect model for four structural factors: (1) skill & grade mix—calculated by dividing the total number of registered nurses by the sum of the total number of registered nurses and other nursing staff reported for the most recent shift; (2) most recent shift

worked (1 = "Early shift"; 2 = "Late shift"; and 3 = "Night shift"); (3) turnover ratio—calculated by dividing the total number of admissions and discharges by the total number of patients on the most recent shift; and (4) somatic ratio—calculated by dividing the total number of patients with somatic diagnoses by the total number of patients on the most recent shift. We extracted the empirical Bayes estimate from the mixed-effect model as a unit-level indicator of nurse staffing.

## Data analysis

Data entry was done by an external data entry service. To detect any systematic errors, 5% of the questionnaires were entered twice. The Match[RN] Psychiatry study team checked for plausibility and consistency of the data.

For the first aim the sample description, mean and standard deviation (SD), as well as median, interquartile range (IQR) and range were calculated for all numeric variables under study. For the categorical variables, total numbers and percentages were calculated.

Using the adjusted staffing model described above plus the variable on temporary nurses' deployment allowed us to fulfill our second aim, *i.e.*, to check whether units frequently deploying temporary nurses have nurse staffing levels similar to those that never or less frequently use them.

To measure the relationships between the deployment of temporary nurses and other permanently-employed nurse outcomes—and therefore to address our third aim—we used linear mixed models. For each outcome variable, we calculated a model adjusting for three individual nurse characteristics: age, gender and employment percentage. On the unit level we included our Empirical Bayes estimates for the same staffing model as in aim two, but without the temporary staffing variable. To control for the structural context, we used a two-step approach, modelling the association first without the leadership variable, then with it. For non-standardized estimates, we reported p-values and 95% confidence intervals. Figure 1 illustrates the conducted analysis.

## Sensitivity analysis

As a relatively large number of 36 units were excluded because of lacking information on temporary nurse deployment endogeneity, *i.e.*, unobserved factors that affect sample selection correlating with the outcome, is a concern. We therefore compared the nurses of the two groups by age, gender and employment percentage using the standardized mean difference (SMD).

## Missing data and linkage procedure

We merged the unit data with the corresponding nurse data set (by their common 'unit code' identifier) and removed all incomplete data. The flow chart in Fig. 2 shows each step taken regarding data linkage and missing data.

For our basic statistical functions we used *RStudio*, version 4.0.5 (2021-03-31) for Windows (*RStudio Team, 2022*) and the *tidyverse* package, version 1.3.1 (*Wickham et al., 2019*). For our linear mixed-effect model, we used the *lmer* package, version 1.1.26 (*Bates*
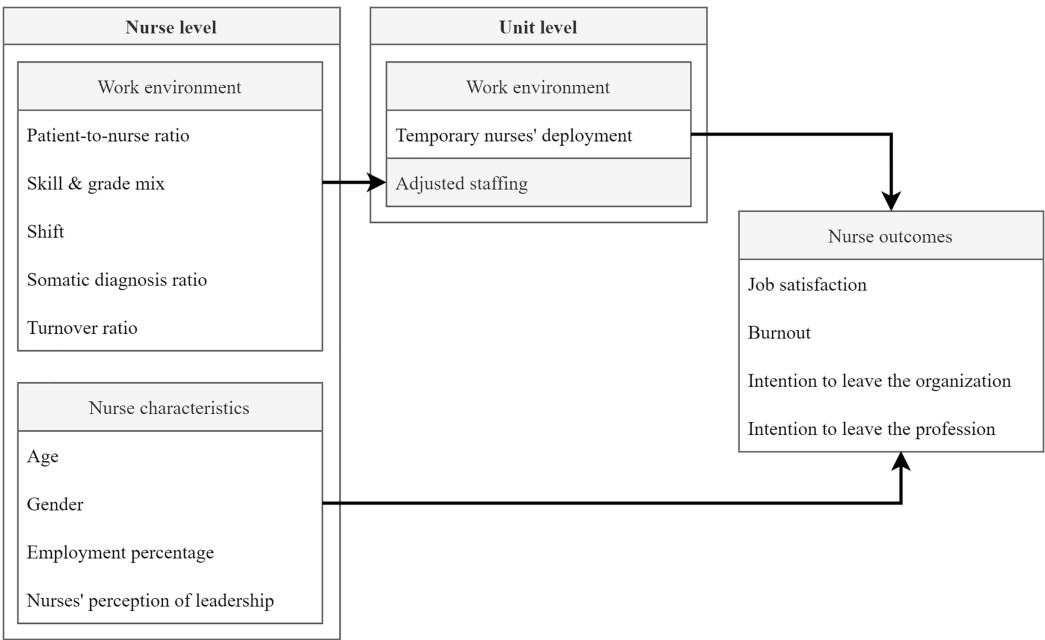

**Figure 1 Conceptual model of the planned analysis.**

*et al., 2015*). For descriptive analysis, we used the *gtsummary* package version 1.4.0 (*Sjoberg et al., 2021*), and the *sjPlot* package version 2.8.10 (*Lüdecke, 2021*). For sensitivity analyses we used the tableone package version 0.12.0 (*Yoshida & Bartel, 2020*). To calculate Cronbach's alpha, we used the *psych* package version 2.1.6 (*Revelle, 2021*). Data and code are available at https://zenodo.org/record/7339714.

# RESULTS

## Sample description

After units with incomplete data were removed, 79 of the initial 115 remained eligible for further analysis. As for the nurse data set, this included responses from 1,012 registered nurses, 879 of whom had responded to all questions regarding the adjusted staffing model. Of these, 689 could be linked to the 79 units eligible for analysis. After 38 more were removed because of remaining missing data of the variables eligible for the final mixed-effect model, 651 remained for the final analysis. Figure 2 presents a flow chart depicting the sequence in which the data sets were screened, including figures for missing data and drop-outs at each stage of data preparation.

Regarding the prevalence of temporary nurse deployment, twenty of the 79 included units' managers (25%) reported frequently deploying temporary nurses; the other 59 (75%) reported occasionally or never deploying them. Further details on unit-level characteristics are provided in Table S1.

Of the 651 nurses on these units who answered all items of interest, the majority (n = 444; 68%) were female. Their average age was 40.8 years (SD = 12.3). Most (n = 288; 44%) were employed between 61% and 95%. Their overall ratings of their units' leadership
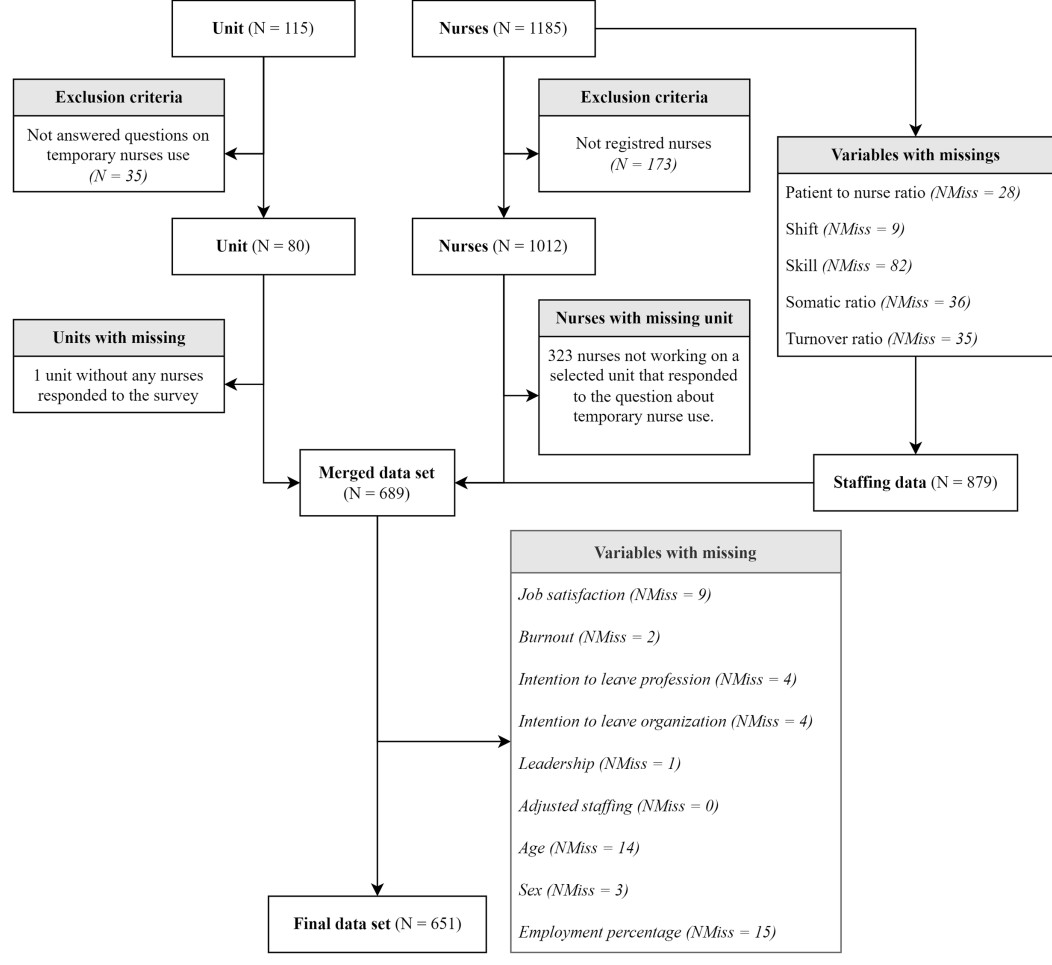

**Figure 2 Flow-chart on missing data and dropouts throughout linkage procedure.** N = number; NMiss = number of missings.

were relatively high (mean: 3.1 out of 4; SD = 0.7). They were generally satisfied with their job (mean = 3.2 out of 4; SD = 0.8), and rarely thought about leaving their organization (mean = 2.0 out of 4; SD = 1.0) or profession (mean = 1.6 out of 4; SD = 0.9). Overall, they reported a moderate incidence of thoughts about burnout (mean = 2.5 out of 5; SD = 0.7). Table 1 summarizes the results of these data's descriptive analysis.

## Staffing model

For an early shift with an average skill & grade mix (71.6%), average somatic ratio (18.8%) and average turnover ratio (12.2%) the average patient to nurse ratio was 7.1 (95% CI [4.77–9.44]). For late shift, the average patient-to-nurse ratio increased by 1.96 patients (95% CI [1.39–2.52]); for night shifts it increased by 8.7 patients (95% CI [8.02–9.57]). Where 10% more patients were admitted or discharged during a shift the average number of patients per nurse decreased by 0.3 (95% CI [−0.4 to −0.1]) and where 10% more patients with somatic diagnoses were present on a shift, the average number of patients per nurse decreased by 0.1 (95% CI [−0.2 to −0.0]). Further details of the adjusted staffing

**Table 1 Descriptive analysis of the nurse sample ($n$ = 651).**

| Variables | N (%) | Mean (SD) | Median (IQR) | Range |
|---|---|---|---|---|
| Job satisfaction | | 3.2 (0.8) | 3.0 (3.0, 4.0) | 1.0, 4.0 |
| Burnout | | 2.5 (0.7) | 2.4 (2.0, 3.0) | 1.0, 4.4 |
| Intention to leave the organization | | 2.0 (1.0) | 2.0 (1.0, 2.0) | 1.0, 5.0 |
| Intention to leave the profession | | 1.6 (0.9) | 1.0 (1.0, 2.0) | 1.0, 5.0 |
| Leadership | | 3.1 (0.7) | 3.2 (2.8, 3.8) | 1.0, 4.0 |
| Age | | 40.8 (12.3) | 40.0 (29.0, 52.0) | 20.0, 66.0 |
| Gender | | | | |
| Female | 444 (68) | | | |
| Male | 207 (32) | | | |
| Employment percentage | | | | |
| <60% | 110 (17) | | | |
| 61–95% | 288 (44) | | | |
| 96–100% | 253 (39) | | | |

Note:

N, number; SD, standard deviation; IQR, interquartile range.

**Table 2 Adjusted staffing model.**

| | Patient-to-nurse ratio | |
|---|---|---|
| *Coefficient* | *Estimates* | *CI (95%)* |
| Intercept | 11.357*** | [10.314–12.400] |
| Skill & grade mix in % | −0.052*** | [−0.064 to −0.039] |
| Late shift | 1.956*** | [1.389–2.522] |
| Night shift | 8.732*** | [7.962–9.501] |
| Somatic diagnoses ratio | −0.012* | [−0.023 to −0.000] |
| Turnover ratio | −0.026*** | [−0.042 to −0.011] |
| Random effects | | |
| $\sigma^2$ | 13.76 | |
| $\tau_{00\ \text{unit\_code}}$ | 1.95 | |
| ICC | 0.12 | |
| $N_{\text{unit\_code}}$ | 114 | |
| Observations | 879 | |
| Marginal $R^2$/Conditional $R^2$ | 0.396/0.471 | |

Notes:

* $p < 0.05$.
*** $p < 0.001$.
CI, confidence interval; $\sigma^2$, residual variance; $\tau_{00\ \text{units}}$, unit level variance; ICC, interclass correlation; N, number; $R^2$, R-squared.

model can be found in Table 2; details of the descriptive analysis of the variables used in the model are provided in Table S2.

Regarding staffing levels, frequent deployment of temporary nurses did not result in significant changes (beta = 0.69; 95% CI [−0.39 to 1.77]). Details are provided in Table S3.

Table 3 Final models with or without leadership and variables related to the four permanently-employed nurses' outcomes.

| Coefficient | Job satisfaction | | Burnout | | Intention to leave the organization | | Intention to leave the profession | |
|---|---|---|---|---|---|---|---|---|
| | Estimates | CI (95%) | Estimates | CI (95%) | Estimates | CI (95%) | Estimates | CI (95%) |
| (Intercept) | 2.859*** | [1.869–3.849] | 2.538*** | [1.807–3.269] | 2.269*** | [1.226–3.313] | 1.916*** | [1.210–2.622] |
| Frequency of temporary nurses' deployment [Frequently] | −0.243* | [−0.454 to −0.031] | 0.254** | [0.098–0.411] | 0.243* | [0.020–0.466] | 0.236** | [0.083–0.388] |
| Random effects | | | | | | | | |
| $\sigma^2$ | 0.63 | | 0.49 | | 0.77 | | 0.71 | |
| $\tau_{00}$ | 0.09 unit_code | | 0.03 unit_code | | 0.09 unit_code | | 0.00 unit_code | |
| ICC | 0.12 | | 0.06 | | 0.10 | | | |
| N | 79 unit_code | | 79 unit_code | | 79 unit_code | | 79 unit_code | |
| Marginal $R^2$/Conditional $R^2$ | 0.028/0.146 | | 0.044/0.100 | | 0.066/0.162 | | 0.039/NA | |
| (Intercept) | 0.803* | [0.101–1.505] | 3.548*** | [2.832–4.264] | 3.927*** | [3.081–4.774] | 2.847*** | [2.124–3.570] |
| Leadership | 0.768*** | [0.687–0.849] | −0.379*** | [−0.460 to −0.297] | −0.609*** | [−0.709 to −0.509] | −0.355*** | [−0.451 to −0.260] |
| Frequency of temporary nurses' deployment [Frequently] | −0.100 | [−0.244 to 0.044] | 0.189* | [0.042–0.336] | 0.132 | [−0.041 to 0.305] | 0.179* | [0.031–0.326] |
| Random effects | | | | | | | | |
| $\sigma^2$ | 0.43 | | 0.43 | | 0.66 | | 0.66 | |
| $\tau_{00}$ | 0.02 unit_code | | 0.03 unit_code | | 0.03 unit_code | | 0.00 unit_code | |
| ICC | 0.05 | | 0.06 | | 0.04 | | | |
| N | 79 unit_code | | 79 unit_code | | 79 unit_code | | 79 unit_code | |
| Marginal $R^2$/Conditional $R^2$ | 0.377/0.407 | | 0.158/0.204 | | 0.247/0.278 | | 0.112/NA | |

Notes:
* $p < 0.05$.
** $p < 0.01$.
*** $p < 0.001$.
CI, confidence interval; $\sigma^2$, residual variance; $\tau$, rank correlation coefficient; ICC, Interclass correlation; $R^2$, R-squared.

## Factors related to permanently-employed nurses' outcomes

Frequent deployment of temporary nurses was associated undesirably with all four of this study's examined permanently-employed nurse outcomes. Specifically, such deployments were associated with a 0.24-point reduction in job satisfaction ratings (maximum: 4; 95% CI [−0.45 to −0.03]; mean = 3.2), a 0.25-point increase in burnout ratings (maximum: 5; 95% CI [0.1–0.4]; mean = 2.5), a 0.24-point higher intention to leave the organization (maximum: 5; 95% CI [0.02–0.47]; mean = 2.0) and a 0.24-point higher intention to leave the profession (maximum: 5; 95% CI [0.08–0.4]; mean = 1.6). Controlling for leadership ratings, frequent deployment of temporary nurses was still associated with a 0.18-point higher intention to leave the profession (maximum: 5; 95% CI [0.03–0.33]; mean = 1.6) and a 0.19-point increase in burnout scores (maximum: 5; 95% CI [0.04–0.34]; mean = 2.5). While the coefficients for job satisfaction and intention to leave the organization were not significant but pointed in the same direction. We also observed a positive association between nurse-rated leadership and all studied nurse outcomes (see Table 3). In addition, we observed higher burnout scores among nurses with higher

employment levels; and older nurses indicated slightly lower intention to leave the organization or profession. These and further details of the models with and without leadership can be found in Tables S4 and S5. Sensitivity analysis did show a SMD of 0.022 for nurses' gender, 0.051 for nurses' age and 0.036 for nurses' employment percentage indicate no substantial differences between the stud sample and excluded units. Further details can be found in Table S6.

## DISCUSSION

Our descriptive analysis indicated that one-quarter of the surveyed units frequently deployed temporary nurses, while most reported only occasionally or never doing so. We did not observe any association between frequent deployment of temporary nurses and differences in nurse staffing. Additionally, we observed that, in the studied psychiatric hospitals, nurses working on units that frequent deployed temporary nurses had significantly higher scores regarding burnout and intention to leave the profession. While their overall ratings for job satisfaction and intention to leave the organization were not significantly associated with frequent temporary nurse deployment, the findings suggested a tendency in that direction.

To our knowledge, no currently available studies in psychiatric hospitals are comparable to this one. Bae, Mark & Fried (2010) showed that roughly 30% of units in somatic hospitals frequently deployed temporary nurses. This supports our finding of a 25% prevalence of temporary nurse deployment in psychiatric hospitals. Perhaps most importantly, frequent deployment of temporary nurses was not associated with changes in staffing levels. This suggests that such deployments prevented understaffing (Shin, Park & Bae, 2018).

In contrast to Aiken et al. (2007), our study found associations between frequent deployment of temporary nurses and all four of the studied permanently-employed nurse outcomes for psychiatric hospitals. When controlling for leadership ratings, we still observed associations with burnout and intention to leave the profession. Any apparent effects on job satisfaction and intention to leave the organization were not significant, but still pointed in the same direction. Using differently-specified models, we found associations between temporary nurse deployment and nurse outcomes; however, it remains unclear whether an unfavorable work environment led to the higher prevalence of vacancies on such units, thereby making it necessary to deploy temporary nurses more frequently, or that having to work more often with temporary nurses led to poorer outcomes among permanently-employed nurses. Firm evidence supports the principle that increases in workload and stress increase the likelihood of burnout and intention to leave the profession (López-López et al., 2019; Sasso et al., 2019); but in contrast, permanently-employed nurses have reported increases in their workloads and stress levels when working with temporary nurses (Berg Jansson & Engström, 2017). Increasing our understanding of the underlying mechanisms will require further studies that correlate data on permanently-employed nurses' absences and vacancies with precise figures regarding temporary nurses' deployment.

## STRENGTHS AND LIMITATIONS

This study has several notable strengths. First, to our knowledge, it is the first to use a multicenter sample of psychiatric hospitals to examine the association between temporary nurses' deployments and permanently-employed nurses' outcomes. Second, we used an unusually accurate approach—one that takes differences in shifts and patient populations into account—to assess nurse staffing levels. Third, examining the relationship between frequent deployment of temporary nurses and nurse staffing levels allowed us to assess whether the temporary nurses achieved their primary goal of managed nurse workloads.

In spite of these strengths, the study also has certain limitations. First, as with any cross-sectional study design, the results allow no inference of causality. Therefore, we were not able to fully assess temporary nurses' impacts on permanently-employed nurses' outcomes. Second, despite a large sample size, many respondents failed to provide all the necessary information. This was particularly problematic regarding unit managers. When their unit-level data were linked with nurse data from the same unit, missed items on the frequency of temporary nurse deployment left many units' nurse-level data ineligible for further analysis.

## CONCLUSIONS

This is the first study to examine whether the deployment of temporary nurses is related to permanently-employed nurses' outcomes in psychiatric hospitals. Temporary nurses appear to counteract the challenge of understaffing; however, their possible roles regarding negative outcomes in permanently-employed nurses remains unclear. The studied nurse outcomes are major drivers of the ongoing and growing problem of understaffing, leading to increased temporary nurse deployments. As it remains unclear whether poor nurse staffing, work environment characteristics or the deployment of temporary nurses causes the outcomes of interest, further studies will be necessary to investigate both the effects temporary nurse deployments have on permanently-employed nurse outcomes and the mechanisms that drive them.

The monetary cost of deploying temporary nurses is quite high, but clear. Until the full non-financial costs of deploying temporary nurses are equally clear, we recommend that unit managers avoid the deployment of temporary nurses if possible. That is, wherever possible, nurse managers should substitute absences and vacancies either from their own or from other units in the same institution.

## ACKNOWLEDGEMENTS

We thank Dr. Karin Zimmermann, Stefan Mitterer, Alisa Cantarero Fernandez, and Michael Ketzer from the University of Basel's Institute of Nursing Science for their helpful feedback throughout the process of writing this report. In addition, we thank Chris Shultis for his language editing.

### Funding

The Match$^{RN}$ Psychiatry study was funded by the participating hospitals and by the Association of Nursing Directors of Psychiatric Institutions in Switzerland (VPPS). This analysis was also supported by the Swiss National Foundation (SNF) as part of Switzerland's National Research Program (NRP) 77 (Project 187433). The funders had no role in study design, data collection and analysis, decision to publish, or preparation of the manuscript.

### Grant Disclosures

The following grant information was disclosed by the authors:
Association of Nursing Directors of Psychiatric Institutions in Switzerland (VPPS).
Swiss National Foundation (SNF).
Switzerland's National Research Program (NRP) 77: 187433.

### Competing Interests

Michael Simon is an Academic Editor for PeerJ.

### Author Contributions

- Leonel Oliveira analyzed the data, prepared figures and/or tables, authored or reviewed drafts of the article, and approved the final draft.
- Beatrice Gehri conceived and designed the experiments, performed the experiments, authored or reviewed drafts of the article, and approved the final draft.
- Michael Simon conceived and designed the experiments, authored or reviewed drafts of the article, and approved the final draft.

### Human Ethics

The following information was supplied relating to ethical approvals (*i.e.*, approving body and any reference numbers):

The study was reviewed by the responsible ethics committee (Ethics Committee Northwestern and Central Switzerland) and received a waiver as our study was outside the ethics committee's jurisdiction (Project ID: Req-2019-00589) (*Gehri et al., 2021*).

### Data Availability

The data and code are available at Zenodo: Leonel Oliveira, Beatrice Gehri, & Michael Simon. (2023). The deployment of temporary nurses and its association with permanent nurses' outcomes in Swiss psychiatric hospitals: A secondary analysis. (1.1) [Data set]. Zenodo. https://doi.org/10.5281/zenodo.7750189.

### Supplemental Information

Supplemental information for this article can be found online at http://dx.doi.org/10.7717/peerj.15300#supplemental-information.

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
