# Peer review of "The deployment of temporary nurses and its association with permanently-employed nurses’ outcomes in psychiatric hospitals: a secondary analysis"

_PeerJ, doi:10.7717/peerj.15300_

## Round 0.1 · original submission · Minor Revisions

Dear Author,

Please address the comments from the reviewers carefully.

Thank you,

Reviewer 1 ·

Basic reporting

no comment

Experimental design

please, find attached.

Need some explanation about the experimental design.

Validity of the findings

no comment

Additional comments

It is interesting study.

Annotated reviews are not available for download in order to protect the identity of reviewers who chose to remain anonymous.

Reviewer 2 ·

Basic reporting

The authors has explained concept of temporary nurse clearly including research problem.
In line 107:
please add supporting data to support the claim that psychiatric hospital has a better planning staffing

Line 117:
as the research variable, the definition of "nurses staffing level" should be globally understand. Thus, we recommend add citation to confirm this term.

Experimental design

Design and setting is clear, including variables and measurement technique

Validity of the findings

we recommend authors report data consistently (n, %) for categorical and (mean, SD) for numerical

Additional comments

To much words and pages, we recommend to concise this manuscript

Reviewer 3 ·

Basic reporting

(1) The English language used in this article is good.
(2) The intro & background is sufficient to demonstrate how this work fits into the broader field of knowledge, though it can be better motivated.
(3) Structure conforms to PeerJ standards and discipline norm.
(4) Raw data has been supplied.
(5) The work is not to test any hypotheses. It's self-contained with relevant results.

Experimental design

(1) This research is within aims and scope of the journal.
(2) The research question is well defined, relevant & meaningful. It is clear enough how the research fills an identified knowledge gap.
(3) The investigation is rigorous enough.
(4) The methods described is with sufficient detail & information to replicate.

Validity of the findings

(1) The reliability of the findings is good.
(2) All underlying data have been provided in a way which we can replicate. The results are statistically sound and controlled. There is no robustness check section in this paper.
(3) Conclusions are well stated. They are linked to original research question and limited to supporting results.

Additional comments

This is an interesting and well-written paper. I have the following additional comments:
(1) The authors should clearly explain why a study focused on psychiatric hospitals is meaningful. In the current version, the authors only highlighted the differences between psychiatric hospitals and acute somatic hospitals.
(2) The authors should add one section to conduct robustness checks. For cross-sectional data, there are still several methods such as Heckman selection model to deal with some endogeneity problems.
(3) The authors should carefully check the tables to make sure no typos. For example, in Table 3, the number of observations is not included.

---

## Round 0.2 · accepted · Accept

Thank you for addressing all of the reviewer comments. This is a well-written manuscript that addresses an important topic. I have assessed this revision without re-inviting the reviewers as all comments have been adequately addressed and I believe this study is ready for publication.